# GROOViST: A Metric for Grounding Objects in Visual Storytelling

**Aditya K Surikuchi**
University of Amsterdam
a.k.surikuchi@uva.nl

**Sandro Pezzelle, Raquel Fernández**
ILLC, University of Amsterdam
{s.pezzelle, raquel.fernandez}@uva.nl

## Abstract

A proper evaluation of stories generated for a sequence of images—the task commonly referred to as visual storytelling—must consider multiple aspects, such as coherence, grammatical correctness, and visual grounding. In this work, we focus on evaluating the degree of grounding, that is, the extent to which a story is about the entities shown in the images. We analyze current metrics, both designed for this purpose and for general vision-text alignment. Given their observed shortcomings, we propose a novel evaluation tool, GROOViST, that accounts for cross-modal dependencies, *temporal misalignments* (the fact that the order in which entities appear in the story and the image sequence may not match), and human intuitions on visual grounding. An additional advantage of GROOViST is its modular design, where the contribution of each component can be assessed and interpreted individually.

## 1 Introduction

Generating a textual story that is plausible given a sequence of images is a challenging task involving aspects such as cross-modal interactions, temporal dependencies between linguistic and visual content, and causal reasoning. In the language-and-vision community, Huang et al. (2016) operationalized the task and released the Visual Storytelling Dataset (VIST), a collection of English stories created by speakers on top of 5-image visual sequences. Several models have been proposed for the task of generating plausible stories for a given sequence, ranging from RNNs (Kim et al., 2018) to Transformers, trained either end-to-end or leveraging additional knowledge-graphs (Chen et al., 2021).

Evaluating the quality of the automatically generated stories is extremely difficult: Given the creative nature of the task (many stories could be sensible for a given image sequence), reference-based metrics like METEOR (Banerjee and Lavie, 2005) or CIDEr (Vedantam et al., 2015) are not

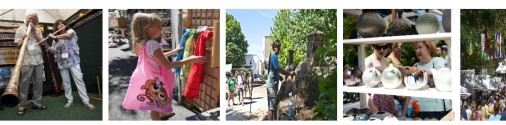

1) there was lots to see and do at the festival , including listening to unusual instruments . 2) many stalls had handmade clothing and one even had dresses specifically for little girls . 3) as part of the festival grounds , there were also numerous sculptures that one could touch . 4) many stalls were adorned with handmade glass bottles . 5) by midday thousands were in attendance , the biggest turn out yet !

Figure 1: One story and corresponding image sequence from the VIST dataset. Noun phrases in green contribute positively to the grounding score by GROOViST; those in red contribute negatively. The GROOViST score for this sample is $0.855$, i.e., our metric considers it as well-grounded (within range: $[-1, 1]$). Best viewed in color.

appropriate—they indeed poorly correlate with human judgments (Wang et al., 2018). Moreover, a proper evaluation must consider multiple aspects, such as coherence, grammaticality and, importantly, visual grounding. Yet, most evaluation metrics proposed specifically for visual storytelling do not consider the images at all (Hu et al., 2020).

In this paper, we focus on evaluating a story's degree of grounding, that is, the extent to which a story is about the entities shown in the images. To the best of our knowledge, there is only one metric proposed to date for evaluating grounding in visual storytelling, the Visual Grounding scorer (RoViST-VG) by Wang et al. (2022). We carry out an extensive analysis of this metric and reveal that it has critical shortcomings. To overcome this, we propose a novel, modular evaluation tool, which we name GROOViST (*grounding objects in visual storytelling*). We show that GROOViST is robust to temporal misalignments, correlated with human intuitions about grounding, and easy to interpret. Our code is available at: https://github.com/akskuchi/groovist

## 2 Analyses of Existing Metrics

To assess the level of visual grounding of a story in visual storytelling, Wang et al. (2022) proposed

RoViST-VG. This metric is the output of a model pre-trained on the Flickr30K Entities dataset (Plummer et al., 2015) to learn the relationships between the nouns in a story and the regions of an image in a contrastive learning regime. For a given <image-sequence, story> pair, RoViST-VG extracts: from each image, the bounding boxes and corresponding visual features of its 10 most salient regions, using FasterRCNN (Ren et al., 2015); from the story, the GloVe (Pennington et al., 2014) representations of each noun in it. The pre-trained model receives these extracted embeddings (from GLoVe and FasterRCNN) and returns the final representations $T$ and $I$, respectively. The grounding score is then calculated using Eq. (1) as the maximum cosine similarity between $T$ and $I$, weighted by inverse document frequencies ($idf$) of the nouns.[1]

$$\text{RoViST-VG} = \log \sum_{i=1}^{|T_e|} \exp(idf(T_i) \max_{I_{e,j} \in I_e} (\cos(T_{e,i}, I_{e,j}))) \quad (1)$$

To analyze the suitability of RoViST-VG, we compare it to CLIPScore (Hessel et al., 2021). CLIPScore has not been designed to evaluate visual storytelling. Here, we use it to score each image-sentence pair independently in a story sequence. This approach is not ideal as it cannot capture temporal misalignments between a text and a visual content (e.g., an early sentence may be '*they were getting ready to go to the circus*' but the circus may only appear later). However, since CLIPScore has been designed for general vision-text alignment, we expect it to be reasonably effective at capturing visual grounding at the image-sentence level. It corresponds to the cosine similarity between CLIP's (Radford et al., 2021) representations of a sentence **c** and an image **v** (with 2.5 as re-scaling factor).

Next we explore how good the above metrics are at capturing grounding in visual storytelling data.

## 2.1 Grounding in visual storytelling datasets

We analyze the scores assigned by these metrics to the stories in three visual storytelling datasets: (1) VIST (Huang et al., 2016), that comprises sequences of five natural images (from Flickr) and corresponding five-sentence stories; (2) AESOP (Ravi et al., 2021), that includes sequences of three synthetic images (created using entities from Abstract Scenes; Zitnick and Parikh, 2013) and corresponding three-paragraph long stories; (3) VWP (Hong et al., 2023), which comprises sequences

---

[1]More details on RoViST-VG are provided in Appendix C.

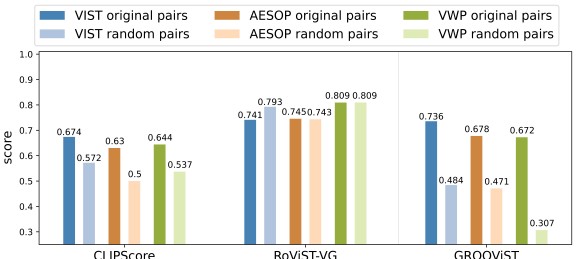

Figure 2: RoViST-VG does not exhibit the expected pattern: it does not assign lower scores to the random <images, story> pairs. In contrast, this is the case for CLIPScore and our proposed GROOViST metric.

of movie shots, each including 5-10 images with corresponding sentences that make up their stories.

We compute RoViST-VG and CLIPScore on the original <image sequence, story> pairs in the test splits of these datasets,[2] and compare these scores to the ones obtained on a *random* setting where each image sequence is paired with five random stories (from the corresponding dataset); among these, we consider the pair that receives the highest score. We expect a metric that properly captures visual grounding to assign higher scores to the original stories than to the randomly paired stories.

Figure 2 shows the average scores of the metrics in both settings. Surprisingly, RoViST-VG scores are not higher in the original setting than in the random setting. In fact, on VIST, the random <image sequence, story> pairs receive higher RoViST-VG scores than the original ones. In contrast, CLIP-Score follows the expected pattern.

## 2.2 Correlation with Flickr8k-Expert ratings

We assess the ability of the two metrics to capture general image-caption grounding using Flickr8k-Expert (Hodosh et al., 2013), a publicly available dataset with human ratings for image-caption pairs. In particular, we consider the subset of 3391 samples where all three annotators agree.[3] CLIPScore is designed for this purpose and is therefore well-suited for the task. RoViST-VG is not meant for measuring image-caption grounding, although it should align with human ratings to some extent, given its purpose and pre-training. However, as we can see in Table 1, RoViST-VG shows no correlation with human ratings—while CLIPScore does.

---

[2]5055 samples for VIST and 991 for AESOP. Due to the lack of a separate test split for VWP, we considered all 13843 samples in the dataset.

[3]Human annotators rated captions on a scale of 1 to 4.

|  | RoViST-VG | CLIPScore | GROOViST |
|---|---|---|---|
| $\tau_c$ (p-value) | 0.019 (0.125) | 0.556 (0.0) | 0.414 (0.0) |

Table 1: Correlation (Kendall $\tau_c$) between Flickr8k-Expert human ratings and the automatic metrics.

## 3 GROOViST

Our analyses showed that RoViST-VG has some important limitations as a metric for assessing the degree of visual grounding—both in stories and image captions. To overcome these issues, we propose GROOViST, a modular metric consisting of various components informed by insights from both CLIPScore and RoViST-VG. These are:

**Noun phrase (NP) extraction** We process the story and extract all the NPs;[4] this is similar to RoViST-VG but better because RoViST-VG only considers nouns and fails to handle compounds such as *'parking lot'*. Additionally, focusing on NPs allows for the contribution of accompanying adjectives (e.g., *'silly faces'*).

**Vision-language alignment** We compute alignment scores between all the extracted bounding boxes and NPs and select the highest score for each NP. This step is similar to RoViST-VG but, instead of training a dedicated model, we use the off-the-shelf CLIP (Radford et al., 2021) model.

**Penalizing poorly grounded NPs** The previous steps result in a positive score for all the NPs in a story. Yet, some may in fact be poorly grounded (i.e., have low visual alignment score). Such NPs, therefore, should contribute *negatively* to the overall degree of grounding of a story. To operationalize this, we select the mean score over all NPs in the entire dataset as a threshold $\theta$ and calculate the distance of each NP's score from $\theta$, assigning negative values to NPs with scores below $\theta$ ($\text{NP}_{neg}$) while retaining the scores of NPs with values above $\theta$ ($\text{NP}_{pos}$).

**Concreteness weighting** RoViST-VG uses inverse document frequencies ($idf$) for weighting the similarity scores of nouns to handle abstract frequent words such as *'time'*. However, we observe that $idf$ weights tend to increase the similarity scores of some less-frequent non-grounded

nouns and decrease the scores of some frequent-and-grounded nouns, adversely affecting the overall score.[5] Hence, after the penalization step, we use word concreteness ratings (Brysbaert et al., 2014) for weighting the resulting scores (instead of $idf$) and capture the fact that concrete NPs are more likely to be visible.[6]

**Normalization** Finally, to obtain the GROOViST score of a story, we aggregate the weighted scores of all its NPs and normalize the sum by the total number of NPs in the story, which results in a value unaffected by story length (or more precisely, by the number of NPs in it):

$$\left( \sum_{i=1}^{n} \text{NP}_{pos_i} + \sum_{i=1}^{m} \text{NP}_{neg_i} \right) / (n + m) \quad (2)$$

where $n$ and $m$ are the number of NPs with positive and negative scores, respectively. See Figure 1 for how this facilitates interpretability. The pseudo-code and a working example for GROOViST are provided in Algorithm 2 and Figure 4, respectively. GROOViST scores are unbounded by default, but $\tanh$ can be used to map them to the $[-1, 1]$ range.

## 4 Role of GROOViST Components

We test GROOViST on the same evaluation criteria used in Section 2. From Figure 2 and Table 1, we observe that GROOViST fares well on both evaluation criteria. First, it assigns higher grounding scores to *original* compared to *random* stories. Second, it moderately correlates with human image-caption ratings. This indicates that GROOViST is a more robust metric than RoViST-VG.

To understand the impact of GROOViST's components on the final grounding score, we conduct several experiments by both ablating the components and replacing them with plausible alternatives.

**Ablations** *Penalizing poorly grounded NPs* and *Concreteness weighting* are the two components of GROOViST that can be ablated from the metric.

**Replacements** The *Concreteness weighting* and *Noun phrase (NP) extraction* components of GROOViST can be replaced with *idf* weights and nouns, respectively.

In total, we consider six alternative versions of our metric, which we obtain by applying all possible combinations of ablations and replacements.

---

[4]Using spaCy's English transformer pipeline for chunking: https://spacy.io/models/en#en_core_web_trf

[5]Examples are provided in Appendix A.

[6]98.7% of NPs in the VIST test set contain words for which concreteness ratings are available.

We test these versions on the same evaluation criteria used in Section 2. Table 2 reports how they fare with respect to the two criteria we consider.

| | Criterion 1 | Criterion 2 |
|---|---|---|
| GROOViST | ✓ | ✓ |
| GROOViST (-C) | ↓ | ✓ |
| GROOViST (-P) | × | ↓ |
| GROOViST (-C -P) | × | ✓ |
| GROOViST (-NPs +Ns) | ↓ | ✓ |
| GROOViST (-C +*idf*) | ✓ | ↓ |
| GROOViST (-C +*idf* -NPs +Ns) | ↓ | ↓ |

Table 2: Results of ablating and replacing different components of GROOViST. C and P refer to *Concreteness weighting* and *Penalizing poorly grounded NPs* components respectively. ✓ indicates that the criteria are met; × indicates that the criteria are not met; ↓ for Criterion 1 indicates a deterioration in the ability of the metric to distinguish between original and random stories. ↓ for Criterion 2 indicates a decrease in the correlation of metric scores with Flickr8k-Expert ratings.

We observe that ablating or replacing components from GROOViST results in scores that either do not meet at least one of the criteria or do so to a much lower extent.[7] This is particularly apparent in the metric versions where the *Penalizing poorly grounded NPs* component is ablated, which further confirms its importance. The GROOViST (-C +*idf*) version satisfies Criterion 1, indicating that frequency-based information can be helpful as a heuristic. However, it may result in discrepancies as shown in Appendix A, Figure 4. We consider concreteness to be a more theoretically motivated notion than frequency to capture visual grounding. Its value is apparent with respect to Criterion 2: replacing *Concreteness weighting* with *idf* weighting decreases the correlation of the metric scores with Flickr8k-Expert ratings.

## 5 Evaluation of GROOViST

To further evaluate the extent to which GROOViST captures intuitions on stories' degree of visual grounding, we compare our metric to human judgments. Since no previous work collected human data for this specific purpose, we run a small data collection by asking 5 participants to rate a sample of the VIST data. In particular, we ask participants to provide ratings for 100 randomly sampled VIST <image sequence, story> pairs, using a 4-point Likert-like scale (instructions: *"a score of 4*

*indicates that most aspects mentioned in the story are depicted in the sequence of images"*).[8] We formulate two hypotheses about the strengths and weaknesses of GROOViST and CLIPScore and experimentally test their validity using the human grounding judgments.

### 5.1 Temporal misalignment

Effective metrics for measuring grounding in visual storytelling should account for possible *temporal misalignments* between the visual and textual modality. That is, they should account for the fact that entities that are grounded in an image could be mentioned earlier or later in the story—not necessarily in the corresponding sentence. We hypothesize that GROOViST—since it takes into account the entire story holistically—correlates better with human judgments than CLIPScore on samples with high temporal misalignment. To test this hypothesis, we define *temporal misalignment* $t$ of a sentence$_i$ in a sequence as the number of its NPs matching with visual entities in images ($img_{j \neq i}$) at other positions of the sequence, normalized by the total number of its NPs. The overall temporal misalignment $T$ of a story is then the average of its sentence-level $t$ values:

$$t(\text{sentence}_i) = \frac{\#(\text{NPs matching img}_{j \neq i})}{\#(\text{NPs in sentence}_i)} \quad (3a)$$

$$T(\text{story}) = \sum_{i=1}^{n} t(\text{sentence}_i) \,/\, n \quad (3b)$$

where $n$ is the number of sentences in a story.

We consider a story to have high temporal misalignment if $T \geq 1.0$, i.e., at least as many as the average number of NPs per sentence are misaligned. In the annotated data, $T \in [0.16, 1.53]$ and 18% of the stories exhibit high temporal misalignment, indicating the prevalence of the phenomenon.

As can be seen in Figure 3, our hypothesis is confirmed: GROOViST exhibits a higher correlation with human ratings than CLIPScore on samples with a high $T$, i.e., its scores are overall more aligned with human intuitions when in the presence of temporally misaligned entities. This confirms the ability of GROOViST to handle non-trivial grounding dynamics in a story, different from CLIPScore. At the same time, we notice that CLIPScore achieves a higher correlation than our metric in samples with low $T$, which confirms once again that the former is an effective tool for capturing grounding in well-aligned multimodal data.

---

[7] The resulting values are provided in Appendix E.

[8] Appendix B provides further details.

## 5.2 Proportion of noun phrases

GROOViST builds on noun phrases. As explained above, this has some obvious advantages, e.g., it allows to measure the individual contribution of each NP toward the final score (see Figure 1), but also some possible limitations. For example, we hypothesize that GROOViST scores may be dependent on the number of NPs; for stories where grounding hinges mostly on NPs, we expect GROOViST to be well aligned with human intuitions; less so when it hinges on verbs, for example, in which case CLIPScore may be better. To test this hypothesis, we define *proportion-of-NPs* ($P$) of a story as the fraction of NPs to all the words in the story:

$$P(\text{story}) = \frac{\#(\text{NPs in story})}{\#(\text{all words in story})} \qquad (4)$$

We focus on the subset of <image sequence, story> pairs with high human ratings,[9] to ensure our analysis genuinely explores the role of NPs in well-grounded stories without being influenced by other factors. We then compute $P$ values for these sequences and bin them into two sets—low $P$ and high $P$—using the distribution's mode (0.2325).[10] The high $P$ bin comprises 32.7% of the total number of subset samples.

In Figure 3, we see that our hypothesis is confirmed. GROOViST scores turn out to be very well aligned with human intuitions—and indeed significantly more correlated than CLIPScore—in the high $P$ bin. In contrast, our metric lags behind CLIPScore in the low $P$ bin, though the distance between the metrics is rather small, and the two metrics generally achieve very low correlations. Although the dependency of GROOViST on the proportion of NPs in a story might be seen as a limitation of the metric, we argue that nouns and accompanying phrases tend to offer the most visual information (Wang et al., 2022). As for RoViST-VG, it achieves a very low correlation with human ratings in both analyses, which confirms its flaws.

## 6 Conclusion

We proposed GROOViST, a novel reference-free metric for evaluating *grounding* in visual storytelling, an aspect that surprisingly is often overlooked in this task. We showed that existing metrics have serious shortcomings, and analyzed the strengths and limitations of our proposed metric.

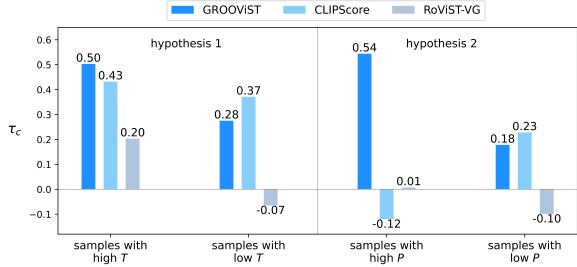

Figure 3: Kendall's $\tau$(variant='c') correlations of all grounding metrics with human scores for temporal misalignment (left) and noun phrase proportion (right).

GROOViST is modular, highly interpretable, and aligned with human intuitions on visual grounding. Preliminary results indicate that GROOViST is a suitable tool for evaluating automatically generated stories. We plan to test this aspect extensively in future work.

## Limitations

In this section, we discuss the limitations specific to our metric and to the general reference-free evaluation paradigm. As discussed in Section 5.2, GROOViST is heavily dependent on noun phrases making it oblivious to other visually informative words, such as verbs. For identifying poorly grounded NPs, GROOViST relies on a threshold value, which is determined based on the dataset of interest. This makes GROOViST vulnerable to the skew of the dataset. Despite our preliminary analysis, GROOViST's evaluation of model-generated stories is yet to be fully tested. Also, in general, reference-free metrics rely on an underlying pre-trained model, which often is stagnant in learning and might require regular fine-tuning for prolonged future relevance. We would also like to underline that throughout this work, we only used and evaluated models trained in the English language text. However, given the modularity of GROOViST, it is possible to switch to models such as multilingual-CLIP (Carlsson et al., 2022).

## Ethics Statement

For collecting human judgments, we recruited participants on a voluntary basis among colleagues of our institution. All data collected for this work is de-identified to ensure the privacy and security of everyone involved. The authors of the VIST dataset (Huang et al., 2016) mention that all images used are CC-licensed.

---

[9]Human rating $\geq 3$ on a scale of 1 to 4.

[10]The same results also hold when using mean and median.

## Acknowledgements

We thank the participants of the human evaluation study and the members of the Dialogue Modelling Group for their vital feedback on the design and experiments. Furthermore, we are grateful to the anonymous EMNLP reviewers for their helpful suggestions and for engaging in fruitful discussions. AKS is supported by the TIMELY project under EU H2020 grant 101017424. RF is supported by the European Research Council (ERC) under the European Union's Horizon 2020 research and innovation programme (grant agreement No. 819455).

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

# Appendix

## A  Text concreteness example

Figure 4 shows that through $idf$ weighting RoViST-VG penalizes the alignment score of the relatively frequent noun *'church'* (0.266). Inversely, $idf$ weighting increases the low alignment score of the abstract and relatively less-frequent noun *'pic'* (0.232). This could have unintended effects on the overall score of the metric, resulting in several discrepancies discussed in Sections 2 and 5.

## B  Human evaluation

We recruited participants on a voluntary basis among colleagues of our institution. We asked the participants for their consent through an informed consent form (see Figure 5). Participants who expressed their consent, were provided access to a scoring web interface with instructions. Each of the participants provided ratings for 100 <image sequence, story> pairs of the VIST data test set on a 4-point Likert-like scale.

## C  RoViST-VG

**Model**  The RoViST-VG model comprises an image-encoder and a text-encoder. The image-encoder encompasses a pre-trained ViT model, an additional linear layer ($W_i$), and a $\tanh$ activation function for obtaining the image representations $I_e$. The text-encoder has a linear layer ($W_t$) and a $\tanh$ activation function for encoding GLoVe vectors into text embeddings $T_e$.

**Pre-training**  The procedure used for pre-training the RoViST-VG model is provided in Algorithm 1.

---

**Algorithm 1** RoViST-VG pre-training

**Input: 1)** A mini-batch of image regions $I_n$ with shape ($m \times 3 \times 224 \times 224$) where $m$ is the batch size. **2)** A mini-batch of matching noun pairs $T_n$ with shape ($m$, 300) where 300 represents the dimensions of the GLoVe vectors. **Output:** Symmetric loss for the mini-batch.

**Initialization:** Pretrained ViT model with linear head for the image encoder, and a single linear layer for the text encoder.

1: $h_n = \text{VisionTransformer}(I_n)$
2: $I_e = \tanh(\mathbf{W}_i \mathbf{h}_n + \mathbf{b}_i)$  ▷ image embeddings; shape=[m,1024]
3: $T_e = \tanh(\mathbf{W}_t \mathbf{T}_n + \mathbf{b}_t)$  ▷ text embeddings; shape=[m,1024]
4: logits = $T_e \times I_e^T$  ▷ shape=[m, m]
5: $I_{sim} = I_e \times I_e^T$  ▷ shape=[m, m]
6: $T_{sim} = T_e \times T_e^T$  ▷ shape=[m, m]
7: labels = $(I_{sim} + T_{sim})/2$  ▷ shape=[m, m]
8: $\mathcal{L}_{image} = \text{cross\_entropy\_loss}(\text{labels}^T, \text{logits}^T)$
9: $\mathcal{L}_{text} = \text{cross\_entropy\_loss}(\text{labels}, \text{logits})$
10: $\mathcal{L}_{symmetric} = (\mathcal{L}_{image} + \mathcal{L}_{text})/2$

---

## D  GROOViST pseudocode

For a given <image sequence, story> pair, the pseudocode in Algorithm 2 outlines the steps involved in computing the GROOViST score.

---

**Algorithm 2** GROOViST

**Input:** Image sequence bounding boxes ($V_i$), corresponding story NPs ($T_i$), concreteness weights ($W_i$), pre-trained CLIP model, score threshold ($\theta$)
**Output:** unbounded GROOViST score $G$
**Steps:**

1: $\text{NP}_{pos}, \text{NP}_{neg} \leftarrow \{\ \}, \{\ \}$
2: **for** $k \leftarrow 1$ to $\#(T_i)$ **do**  ▷ # = number of NPs
3:   $np_e, w \leftarrow \text{CLIP}(T_i[k]), W_i[k]$
4:   $np_s \leftarrow 0.0$  ▷ score of noun phrase $np$
5:   **for each** $v \in V_i$ **do**
6:     $v_e \leftarrow \text{CLIP}(v)$
7:     $np_s \leftarrow \max(np_s, \cos(np_e, v_e))$
8:   **end for**
9:   **if** $np_s \geq \theta$ **then**
10:     $\text{NP}_{pos} \leftarrow np_s \times w$
11:   **else**
12:     $\text{NP}_{neg} \leftarrow -(\theta - np_s) \times w$
13:   **end if**
14: **end for**
15: $G = (\sum \text{NP}_{pos} + \sum \text{NP}_{neg})\ /\ \#(T_i)$

---

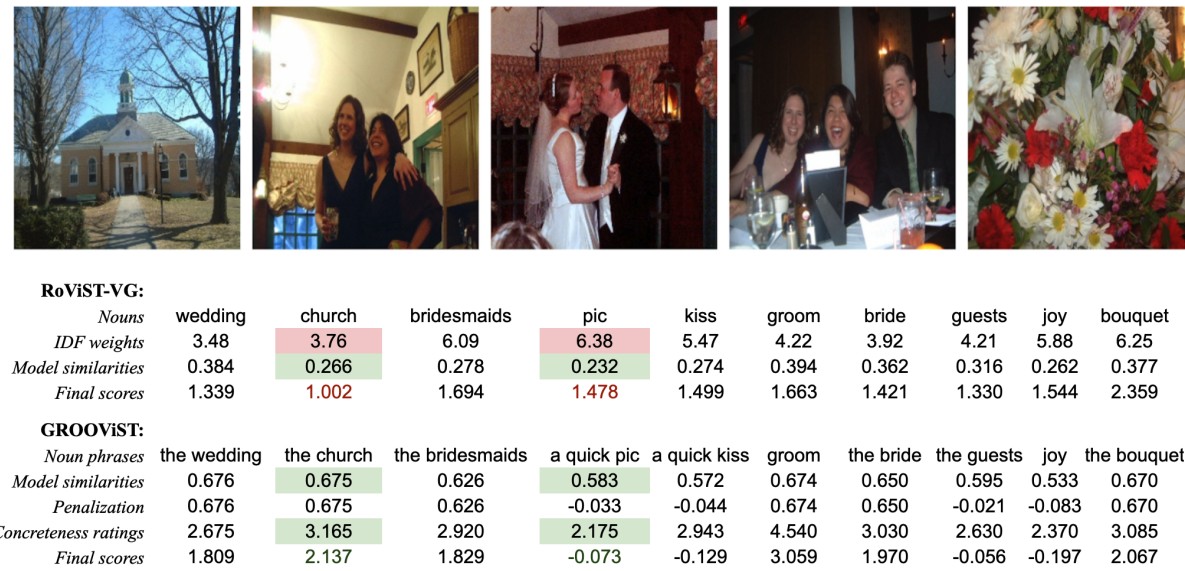

**RoViST-VG:**

| *Nouns* | wedding | church | bridesmaids | pic | kiss | groom | bride | guests | joy | bouquet |
|---|---|---|---|---|---|---|---|---|---|---|
| *IDF weights* | 3.48 | 3.76 | 6.09 | 6.38 | 5.47 | 4.22 | 3.92 | 4.21 | 5.88 | 6.25 |
| *Model similarities* | 0.384 | 0.266 | 0.278 | 0.232 | 0.274 | 0.394 | 0.362 | 0.316 | 0.262 | 0.377 |
| *Final scores* | 1.339 | 1.002 | 1.694 | 1.478 | 1.499 | 1.663 | 1.421 | 1.330 | 1.544 | 2.359 |

**GROOViST:**

| *Noun phrases* | the wedding | the church | the bridesmaids | a quick pic | a quick kiss | groom | the bride | the guests | joy | the bouquet |
|---|---|---|---|---|---|---|---|---|---|---|
| *Model similarities* | 0.676 | 0.675 | 0.626 | 0.583 | 0.572 | 0.674 | 0.650 | 0.595 | 0.533 | 0.670 |
| *Penalization* | 0.676 | 0.675 | 0.626 | -0.033 | -0.044 | 0.674 | 0.650 | -0.021 | -0.083 | 0.670 |
| *Concreteness ratings* | 2.675 | 3.165 | 2.920 | 2.175 | 2.943 | 4.540 | 3.030 | 2.630 | 2.370 | 3.085 |
| *Final scores* | 1.809 | 2.137 | 1.829 | -0.073 | -0.129 | 3.059 | 1.970 | -0.056 | -0.197 | 2.067 |

1) this is the church where the wedding was held . 2) the bridesmaids took a quick pic together . 3) the bride and groom leaned forward for a quick kiss . 4) the guests were overwhelmed with joy . 5) the bouquet was beautiful .

Figure 4: Example showing discrepancies of RoViST-VG *idf* weighting along with GROOViST noun phrase contributions for comparison. The overall normalized GROOViST score for this sample is 0.846.

**Hello!**

In this experiment, we ask you to judge sequences of images and short stories about these sequences. You will be asked to use a numeric score to rate how much the stories are a good fit for the images. This information will help the investigators to devise an automatic evaluation metric to judge stories.

Only the scores you provide and your email address (for identifying your scores) are collected as data.

If necessary, only the scores would be made publicly available for future research.

If you select the "I consent" option, you acknowledge that:

- Your participation is voluntary.

- You are older than 18 years of age.

- You are aware that you may choose to terminate your participation at any time for any reason.

- The data you provide (only the scores), if necessary, will be published and made available for future research.

> I consent

> I do not consent

Thank you for your interest to help our research.

>>

Figure 5: Informed consent form used for recruiting participants.

# E    Ablation and replacement results

|                              | $\Delta$ on VIST | Flickr8k-Expert $\tau_c$ |
|------------------------------|------------------|--------------------------|
| GROOViST                     | 0.413            | 0.414                    |
| GROOViST (-C)                | 0.127            | 0.421                    |
| GROOViST (-P)                | -0.071           | 0.289                    |
| GROOViST (-C -P)             | 0.025            | 0.461                    |
| GROOViST (-NPs +Ns)          | 0.260            | 0.410                    |
| GROOViST (-C +*idf*)         | 0.890            | 0.379                    |
| GROOViST (-C +*idf* -NPs +Ns)| 0.348            | 0.341                    |

Table 3: Results of ablating/replacing components of GROOViST. $\Delta$ refers to the difference between the scores obtained by the original and random <image sequence, story> pairs—the higher the better.