# OpenReview forum: "GROOViST: A Metric for Grounding Objects in Visual Storytelling"
_EMNLP/2023/Conference — EMNLP 2023 Main_

### Official Review · Reviewer_Pc9w · 2023-07-26

**Soundness:** 4

**Excitement:**

4: Strong: This paper deepens the understanding of some phenomenon or lowers the barriers to an existing research direction.

**Paper Topic And Main Contributions:**

This paper proposes a new automatic metric to assess the quality of Visual Storytelling systems. They show that previous metrics used by the community have several weaknesses. Based on those weaknesses, they propose a new metric, which they show to mitigate those weaknesses. Additionally, they show how their metric correlates better with human judgement.

**Questions For The Authors:**

A- You explain your proposed metric in Section 3. When I first read it, I didn't find any specific way to address temporal alignment between sentences and images. However, in Section 4.1 you comment: "We hypothesize that GROOViST—since it takes into account the entire story holistically—correlates better with human judgments than CLIPScore". Don't you think you should mention this explicitly in Section 3? There are some mentions, like in lines 151-152: "We process the story and extract all the NPs", but maybe this should be highlighted better.
B- Regarding Noun Phrase (NP) extraction, starting in line 151, you don't specify how you accomplish the task. What tool(s) are you using for NP extraction?
C- A similar question for Vision-language alignment (line 158): how do you extract object bounding boxes? Faster R-CNN? Another object detector? And how do you use CLIP to compute the similarity of those object-based visual features and NPs? As far as I know, CLIP has been trained for entire images, using different visual backbones trained from scratch (ViT, Resnet...). If you use Faster R-CNN for object detection, I assume you have to use the same visual backbone as CLIP, at least. Please, explain this further.
D- Your analysis shows that CLIPScore may be better for some circumstances (Figure 3, samples with low T and samples with low P). Can you provide a distribution of samples for high/low T and P for the datasets you use? This would be interesting, in order to see whether your metric is better "only for marginal cases" or instead, those are the most frequent cases.

**Reasons To Accept:**

A- The paper is very well motivated.
B- Authors show the weaknesses of previously proposed (and published) metrics. Specially surprising is the bad behavior of RoVIST-VG.
C- The methodology to evaluate automatic metrics for Visual Storytelling is sound and interesting.
D- Their proposed metric is shown to work better than previous metrics, in some important cases.
E- I personally think that research papers about evaluation are very important in our community and this work goes exactly in that direction.

**Reasons To Reject:**

I do not see any (important) reason to reject the paper.

**Reproducibility:**

4: Could mostly reproduce the results, but there may be some variation because of sample variance or minor variations in their interpretation of the protocol or method.

**Reviewer Confidence:**

4: Quite sure. I tried to check the important points carefully. It's unlikely, though conceivable, that I missed something that should affect my ratings.

---

> ### Author Rebuttal · Authors · 2023-08-28
>
> We would like to thank you for your review and questions.
>
> **Responses to Questions:**
>
> A. We agree that this point should also be mentioned earlier in the paper. We will update the camera-ready version accordingly.
>
> B. Thank you for underlining this. We experimented with 2 existing libraries – NLTK pos-tagging-chunking and Spacy noun chunks (based on its English transformer pipeline). While the two libraries lead to comparable performance, upon manual inspection, we found Spacy to be more consistent in identifying phrases. We will release the code for extracting noun-phrases using both libraries, while defaulting to Spacy.
>
> C. We extract bounding boxes using FasterR-CNN. Then, for each bounding box, we extract visual feature representations using the CLIP model. To clarify this further, FasterR-CNN is only used to identify and "crop" the various objects in a given image. Once we have these cropped bounding boxes for each image in a given sequence, we then compute the similarity scores between each NP in the corresponding story and every bounding box of the image-sequence. Finally, we pick the maximum score for every NP.
>
> D. For this analysis, as mentioned at the beginning of Section 4, we asked the participants to provide ratings for 100 randomly sampled VIST <image sequence, story> pairs. Among these samples, $18$% of the stories exhibit high temporal misalignment (more details in Section 4.1). Similarly, $32.7$% of the subset of samples we consider for the P analysis have a high proportion of NPs ($>$ central tendencies of the P distribution – e.g., mode). We agree that expanding and discussing these distribution details would substantiate our findings further. We will profit from the extra page given to camera-ready papers to do so.

---

### Official Review · Reviewer_sXd8 · 2023-08-04

**Soundness:** 3

**Excitement:**

3: Ambivalent: It has merits (e.g., it reports state-of-the-art results, the idea is nice), but there are key weaknesses (e.g., it describes incremental work), and it can significantly benefit from another round of revision. However, I won't object to accepting it if my co-reviewers champion it.

**Paper Topic And Main Contributions:**

The short paper proposes a metric to evaluate temporal misalignments in visual storytelling models. It offers a critical analysis of existing metrics for this task and includes experimental results to support its claims.

**Reasons To Accept:**

- The paper thoroughly examines the limitations of current metrics, providing valuable insights for the field.
- Experimental results demonstrate the effectiveness of the proposed metric.
- The paper is well written, with detailed problem descriptions.


**Reasons To Reject:**

- Limited Generalization: The paper does not adequately address how the proposed metric applies to various kinds of visual storytelling methods. Further clarification is needed in this aspect to ensure broader applicability.
- The paper lacks a detailed experiment on the modularity of the proposed metric.


**Reproducibility:**

4: Could mostly reproduce the results, but there may be some variation because of sample variance or minor variations in their interpretation of the protocol or method.

**Reviewer Confidence:**

3: Pretty sure, but there's a chance I missed something. Although I have a good feel for this area in general, I did not carefully check the paper's details, e.g., the math, experimental design, or novelty.

---

> ### Author Rebuttal · Authors · 2023-08-28
>
> Many thanks for your review.
>
> **Responses for Reasons-to-Reject:**
>
> 1. Thanks for asking us about the generalizability of our metric.
> 	- First, in our paper we showed that GROOViST can be used with (and is suitable for) different visual storytelling datasets, namely, VIST and AESOP – the two visual storytelling datasets available at the submission time. Since a new dataset, Visual Writing Prompts, was recently released by *Hong et al. (2023)*, we tested GROOViST on it. We found that GROOViST works well on this dataset, too, which provides further evidence for the generalizability of our metric. We will report the results of this additional analysis to the updated version of the paper.
> 	- Second, while we developed and validated GROOViST using stories produced by human speakers, we argue that it can be used to assess the degree of grounding of model-generated stories. Indeed, we have preliminary experiments on this, which show the potential of applying GROOViST to this domain. We will discuss this point in the revised version of the paper.
>
> 2. Thanks for raising this important point. Please check our response to Reviewer G7ME for a detailed answer.
>
> References:
> Xudong Hong, Asad Sayeed, Khushboo Mehra, Vera Demberg, Bernt Schiele; Visual Writing Prompts: Character-Grounded Story Generation with Curated Image Sequences. Transactions of the Association for Computational Linguistics 2023; 11 565–581. doi: https://doi.org/10.1162/tacl_a_00553

---

### Official Review · Reviewer_G7ME · 2023-08-10

**Soundness:** 4

**Excitement:**

3: Ambivalent: It has merits (e.g., it reports state-of-the-art results, the idea is nice), but there are key weaknesses (e.g., it describes incremental work), and it can significantly benefit from another round of revision. However, I won't object to accepting it if my co-reviewers champion it.

**Paper Topic And Main Contributions:**

This work introduces a new metric for evaluating grounding in the task of visual storytelling. GROOViST builds upon existing visual grounding metrics for visual storytelling, in the form of RoViST-VG and pre-trained vision-language models, in the form of CLIP (and the associated CLIPScore).

GROOViST brings several improvements over previous metrics, such as considering noun phrases instead of only nouns, computing image-text alignment using CLIP (instead of training a specific model for this task), replacing idf weighting with “concreteness” weighting. Finally, for each NP they assign a score that can also be negative (probably for interpretability reasons), averaging over all these scores to obtain the final value of the metric.

Evaluations on two datasets are performed and a couple of hypotheses about the strengths and weaknesses of the metrics are tested with good results.

**Questions For The Authors:**

A. Attributing negative scores to NPs, seems a convoluted step and is not clear how much improvement does it really bring. Is it purely for interpretability?

B. In Figure 1, you state that “[…] score for this sample is 1.278, i.e., our metric considers it as moderately grounded[…]”. Can you provide more details about the range of the metric? What is a bad score?

C. Why is the normalization equation (eq 2), split into “positive” and “negative” NPs?

D. What is exactly modular about GROOViST?

**Reasons To Accept:**

This paper introduces a new metric for evaluating grounding in generated visual stories. The idea is simple and clean, well presented and the paper is easy to read.

GROOViST builds upon the weaknesses of previous metrics, generating in the end a better, more aligned with human judgment visual storytelling metric.

**Reasons To Reject:**

I strongly believe ablation studies are needed in this work. The authors propose a few improvements, that work (the proposed metric seems to be better than previous metrics), but do not properly evaluate which parts are bringing this improvement and to what extent.
“GROOViST is its modular design, where the contribution of each component can be assessed and interpreted individually” (from abstract). I find this exact interpretation and analysis missing from this work.

How, by how much, in which cases does the step from nouns to noun phrases aids performance? Similar questions should be answered for replacing idf with concreteness weighting.

**Reproducibility:**

4: Could mostly reproduce the results, but there may be some variation because of sample variance or minor variations in their interpretation of the protocol or method.

**Reviewer Confidence:**

3: Pretty sure, but there's a chance I missed something. Although I have a good feel for this area in general, I did not carefully check the paper's details, e.g., the math, experimental design, or novelty.

---

> ### Author Rebuttal · Authors · 2023-08-28
>
> We would like to thank you for your detailed and insightful review.
>
> For clarity reasons, we will start by answering your 4 questions and then address your concerns under Reasons to reject.
>
> **Responses to Questions:**
>
> A. Penalizing poorly grounded NPs based on a factor $\theta$ is an essential step before aggregation. Summing up the individual NP scores without this penalization step would indeed lead to an overall score that is overinfluenced by highly grounded NPs. Given the goal of our metric, this is undesirable. Consider the case where we have two stories, a well-grounded one, and one mentioning some ill-grounded – if not unrelated – NPs. Suppose that the two stories receive the following sets of scores (based on their NPs): $\lbrace 0.63, 0.605, 0.61 \rbrace$ and $\lbrace 0.33, 0.74, 0.8, 0.48, 0.62, 0.72 \rbrace$ for the well-grounded and ill-grounded, respectively. Aggregating the scores without penalizing the NPs responsible for the ill-groundedness would lead to exactly the same GROOViST score for both stories: $1.845/3 = 3.69/6 = 0.615$. This signifies the importance of the penalization step in making the metric scores sensible. As a by-product of this step, our metric is also highly interpretable, as shown in Figures 1 and 4.
>
> B. The range of the metric is unbounded by default. In Fig 1, we consider the score to be moderately grounded based on the distribution of scores of the dataset. We will further clarify this aspect in the final version.
>
> C. Eq. 2 reflects the difference between positive and negative NPs, as explained in our response to question A above.
>
> D. GROOViST builds on 4 components or modules: NP extraction, vision-language alignment, concreteness weighting, and penalizing poorly grounded NPs; followed by a normalization step. All these components are key to the functioning of our metric. At the same time, each of them can be updated or replaced if needed (e.g., CLIP-based vision-alignment scores could be replaced with scores by a different model). As such, our metric is model-agnostic.
>
> **Responses for Reasons-to-Reject:**
>
> 1. We believe we further clarified the modular nature of our metric in our answer to your question D. As for the need of an ablation analysis, this is a very important point and we thank you for raising it. Following your comment, we performed an ablation analysis by removing, one at a time, the modules that – by definition – can be removed from the metric without needing a replacement by some other alternative (e.g., the NPs module cannot be ablated, it can only be replaced, e.g., by nouns), namely, concreteness weighting and the penalization of poorly grounded NPs. By evaluating the ablated versions of the metric following the steps described in section 2, we found that neither of the two meet the desired criteria of differentiating original <image, story> pairs from random pairs. We observed the same when jointly ablating the two modules. This shows that both these modules are essential for the calculation of GROOViST. Profiting from the additional page, we will include this analysis and its results in the final version of the paper.
>
> 2. The choice of using NPs over Ns is motivated by linguistic reasons. In fact, using simple Ns over NPs splits compound nouns such as “*parking lot*” or “*post office*” into two separate nouns, which is clearly undesirable. However, given the modular nature of our metric, we can still replace NPs with Ns and test the effect of it. We tested so by means of an analysis similar to the one reported above. As expected, this deteriorates the ability of the metric to distinguish between original and random stories, which validates our theoretical motivation.
> As for the use of concreteness weighting, this is motivated in Section 3. We observed that IDF has several flaws when used in a metric that aims at evaluating the degree of grounding of a text. Moreover, it is not informative of the degree of abstractness/concreteness of words, as we also highlight in Fig. 4 in the Appendix. However, following your comment, we did test the effect of replacing concreteness weighting with idf in our metric (using the IDF scores of the NPs). In this case, we did not observe a deterioration of the metric with respect to the criteria described in Section 2. We conjecture that this may be due to the effect of the other modules, making IDF’s shortcomings less impactful. We will report this analysis in the paper and discuss its implications.

---

### Meta-Review · Area_Chair_R7Yj · 2023-09-19

**Recommendation:** 4

**Metareview:**

This paper proposes a new evaluation metric for visual storytelling, focusing on how references to objects and events are grounded in the sequence of images on which the story generation was conditioned. The proposed metric is used to evaluate methods applied to two visual storytelling datasets (including a third dataset in the author rebuttal). Evaluation shows that the metric correlates with human judgments of temporal and visual grounding.

---

### Decision · Program_Chairs · 2023-10-07

**Decision:**

Accept-Main

**Comment:**

This paper proposes a new evaluation metric for visual storytelling, focusing on how references to objects and events are grounded in the sequence of images on which the story generation was conditioned. The proposed metric is used to evaluate methods applied to two visual storytelling datasets (including a third dataset in the author rebuttal). Evaluation shows that the metric correlates with human judgments of temporal and visual grounding.